# Antibacterial Activity against Clinical Strains of a Natural Polyphenolic Extract from Albariño White Grape Marc

**DOI:** 10.3390/ph16070950

**Published:** 2023-07-01

**Authors:** Tamara Manso, Marta Lores, José Luis R. Rama, Rosa-Antía Villarino, Lorena G. Calvo, Aly Castillo, María Celeiro, Trinidad de Miguel

**Affiliations:** 1Complejo Hospitalario Universitario de Ferrol, E-15405 Ferrol, Spain; tamara.manso.gomez@sergas.es; 2Laboratory of Research and Development of Analytical Solutions (LIDSA), Department of Analytical Chemistry, Nutrition and Food Science, Universidade de Santiago de Compostela, E-15782 Santiago de Compostela, Spain; marta.lores@usc.es (M.L.); alyjesus.castillo.zamora@usc.es (A.C.); maria.celeiro.montero@usc.es (M.C.); 3i-Grape Laboratory, Emprendia, Campus Vida, E-15782 Santiago de Compostela, Spain; 4Department of Microbiology and Parasitology, Universidade de Santiago de Compostela, E-15782 Santiago de Compostela, Spain; joseluis.rodriguez.rama@usc.es (J.L.R.R.); rosaantia.villarino@rai.usc.es (R.-A.V.); lorena.gomez.calvo@rai.usc.gal (L.G.C.); 5CRETUS, Department of Analytical Chemistry, Nutrition and Food Science, Universidade de Santiago de Compostela, E-15782 Santiago de Compostela, Spain

**Keywords:** antibacterial activity, natural extract, white grape marc, polyphenols, clinical strain, antimicrobial resistance

## Abstract

Infections caused by multidrug-resistant bacteria are becoming increasingly frequent and sometimes difficult to treat due to the limited number of antibiotics active against them. In addition, they can spread between countries and/or continents, which is a problem of great relevance worldwide. It is, therefore, urgent to find alternatives to treat infections caused by multidrug-resistant bacteria. This study aimed at exploring a possible therapeutic alternative in the fight against antibiotic resistance. Based on the known antibacterial capacity of polyphenols, we tested the antimicrobial activity of a polyphenolic extract of Albariño white grape marc on clinical strains since research on such bacteria has been very scarce until now. First, the extract was obtained using a medium-scale ambient temperature (MSAT) system, which is an efficient and sustainable extractive method. The determinations of the polyphenolic content of the extract and its antioxidant capacity showed good results. Using chromatographic and mass spectrometric tools, 13 remarkable polyphenols were detected in the extract. The antibacterial activity of our grape marc extract against nineteen clinical strain isolates, some of which are multidrug-resistant, was evaluated by means of the calculation of half of the maximum inhibitory concentration (IC50) and the value of the minimum bactericidal concentrations (MBCs). In conclusion, the extract showed effectiveness against all clinical strains tested, regardless of their level of antibiotic resistance, and shows promise in the fight against antibiotic resistance.

## 1. Introduction

The discovery of penicillin in 1929 [1] represented a major advance in the fight against infectious agents, significantly reducing the number of infections, as well as the deaths associated with infectious diseases. This was followed by the discovery and improvement of other antibiotics, which gave rise to the wide variety of such molecules with different targets of action available today [2].

Due to natural processes and the selective pressure exerted by antibiotics, bacteria can become resistant to these drugs, and this process is amplified by antibiotics’ widespread and sometimes incorrect and uncontrolled use in agriculture and human and animal health. According to the European Centre for Disease Prevention and Control (ECDC), Spain is one of the European countries with the highest use of systemic antibiotics in the community and hospitals, being ranked sixth in 2021 [3].

Infections caused by resistant microorganisms have few therapeutic alternatives. This has serious consequences for health (increased morbidity and mortality, higher cost of health care), and hinders important medical advances, such as transplants, cancer treatments and surgeries [4]. Therefore, the emergence and spread of multidrug resistances (MDRs), which refers to bacteria resistant to at least three families of antibiotics [5], as well as the lack of alternative drugs to combat them, is a serious threat to health and a global problem that requires coordinated solutions [4,6,7,8]. The United Nations [9], the European Antimicrobial Resistance Surveillance Network (EARS-Net) [10] and several European countries aim at fighting antimicrobial resistance (AMR) using their national plans [4,11,12,13].

The number of MDR strains of clinically relevant microorganisms is increasing alarmingly. The World Health Organization (WHO) focuses on MDR Gram-negative bacteria [14], highlighting carbapenem-resistant *Pseudomonas aeruginosa* and *Acinetobacter baumannii* and some Enterobacteria resistant to third-generation cephalosporins and carbapenems. The term “ESKAPE” refers to six priority pathogens with increased resistance and virulence: *Enterococcus faecium*, *Staphylococcus aureus*, *Klebsiella pneumoniae*, *A. baumannii*, *P. aeruginosa* and *Enterobacter* spp. [15]. These bacteria cause most nosocomial infections and can escape the action of antibiotics [16].

Because of the aforementioned threat to global health, which is spreading rapidly around the world, it is of utmost importance to find proposals to tackle AMRs. Some alternatives that could help to solve this problem are based on controlling the use and consumption of antibiotics, monitoring the development of bacterial resistance, developing new antibiotics, or finding alternatives to prevent and treat infections [4].

The development of new antibiotics in recent decades is scarce due to the low profitability for pharmaceutical companies. Of the eight drugs approved in 2017, only two represent a new chemical family, while the other six come from already-known families [17]. These antibiotics focus their action on multiresistant Gram-negative bacilli, and are especially active against multiresistant Enterobacteria [18]. The development of alternative methods to treat bacterial infections is vitally important. Thus, among the options under study, in this work, we focused on secondary metabolites derived from plants called polyphenols, in particular, those present in white grapes. Although this fruit is mainly used to produce wine, this process generates a large quantity of by-products that turn out to be rich in bioactive compounds, such as polyphenols, which are of great interest at the industrial level [19,20]. Around 13% of the total amount of the grapes processed to produce wine become a by-product after pressing known as the grape marc. The main bioactive compounds of grapes are polyphenols, which represent 70% of the total bioactive compounds of the fruit [21]. A large amount of the initial quantity of polyphenols of white grapes remains in the grape marc, turning it into a by-product with an enormous bioactive potential [22].

The basic structure of polyphenols consists of a benzene ring with at least one hydroxyl group [23]. This ring may also contain different substitutes, which may be combined with sugars. Therefore, the structure of polyphenols is very diverse, ranging from very simple molecules to complex polymeric structures. Their classification is based on the number of carbon atoms in their basic molecular skeleton [24]. Table 1 gives the main classes, subclasses and representatives of polyphenols found in white grapes.

There is extensive knowledge of the antimicrobial and antioxidant properties of polyphenols and polyphenol-rich extracts, but there is a lack of studies involving clinical isolates, as shown by a recent review by Manso et al. [25]. Only one of the studies reviewed in this article used grape extracts [26].

Our search for literature about the antibacterial activity of white grape extracts resulted in a review of 13 studies, of which only 4 employed clinical strains. There is therefore a need for more research in this field. Taking this into account, the aim of this study was to use a grape extract from Albariño white grape marc to evaluate its activity against clinical strains. In addition to the fact that the antimicrobial properties of these by-products, which are related to their polyphenolic profile, could provide an alternative to the problem of AMR, the re-use of waste generated in the wine industry would be very beneficial from environmental and economic points of view.

## 2. Results

### 2.1. Sensitivity Study of the Isolate Strains

Several strains isolated from clinical samples were selected to evaluate the antibacterial activity of the polyphenolic extract of Albariño white grape marc. These pathogenic strains, their acquired resistance mechanism (when detected), the clinical sample from which they were isolated and the culture medium used for the initial recovery are shown in Table 2 and Table 3. The samples were processed in the microbiology laboratories of two medical centers: Hospital Público da Mariña (Burela, Lugo, Spain) and Complejo Hospitalario Universitario de Ferrol (Ferrol, A Coruña, Spain), hereinafter HPM and CHUF, respectively. The equipment used to identify and carry out the sensitivity study of each bacterium is cited in the Section 4. 

Table 4 and Table 5 show the antibiotics tested against each clinical strain, as well as the interpretation of the tests according to the EUCAST (European Committee on Antimicrobial Susceptibility Testing) guideline 2022, except for *B. cepacia*, for which the CLSI (Clinical and Laboratory Standards Institute) guideline 2022 was used since the EUCAST guideline has not determined cut-off points for it.

Acquired resistance was detected in Gram-negative bacilli: *P. fluorescens*, *K. pneumoniae*, *P. mirabilis*, *E. coli*, *C. freundii* and *S. enteritidis*, and Gram-positive cocci *S. aureus* and *S. epidermidis*. The carbapenemase-producing enzymes VIM and oxa-48 were detected in *P. fluorescens* and *K. pneumoniae* strains using polymerase chain reaction (PCR). PCR was also performed on *P. aeruginosa* and *E. cloacae* strains due to their resistance to carbapenems, obtaining negative results. Detection of the extended-spectrum beta-lactamase-producing enzyme (ESBL) in the strains of *P. mirabilis*, *E. coli*, *C. freundii* and *S. enteritidis* was carried out using antibiotic discs (cephalosporins, monobactams and clavulanic acid) [27] and determination of methicillin resistance in *S. aureus* and *S. epidermidis* was verified using a cefoxitin disc [28]. All these bacteria, except *S. aureus*, fell within the Magiorakos definition of MDR [5].

### 2.2. Chemical–Analytical Characterization of the Extract

The extract of grape marc showed a dark brown color, a pH of 3.94 and a density of 1.04785 g/mL. It contained 10.02% solids and 73.59% water. The humidity of the grape marc employed as raw material was 65.37% and this value was used to express data on a dry weight basis.

The total polyphenol index (TPI) was obtained using the Folin–Ciocalteu method and was found to be 8275 ± 744 mgGAE/L (GAE: gallic acid equivalent). The result of the antioxidant activity of the extract was 51.7 ± 4.1 mmolTRE/L using the 2,2-diphenyl-1-picrylhydrazyl (DPPH) reagent and 71.3 ± 5.7 mmolTRE/L (TRE: Trolox equivalent) with 2,2′-azinobis (3-ethylbenzothiazoline)-6-sulfonic (ABTS), while the IC50 values were 61.2 and 60.07 μmolGAE/L, respectively. The IC50 is the amount of extract that is able to neutralize 50% of the free radicals (DPPH or ABTS) initially present in the solution.

The chromatographic profile (total ion chromatogram) of the isovolumetric ethyl lactate–water extract and the SRM-extracted chromatograms for the target analytes are shown below (Figure 1 and Figure 2). The corresponding individual polyphenol concentrations are depicted in both Figure 3 and Table 6.

### 2.3. In Vitro Methods for the Evaluation of the Antibacterial Activity of the Extract

The antibacterial activity of the extract against the tested bacteria was assessed by calculating the minimum bactericidal concentration (MBC, i.e., the lowest concentration of an antimicrobial agent capable of removing 99.9% of an organism in vitro) [29] and half the maximum inhibitory concentration (IC50, i.e., the concentration of an antimicrobial agent capable of reducing the bacterial concentration to 50% of the initial concentration).

The values of IC50 were calculated using the free online Quest Graph IC50 Calculator (AAT Bioquest 2022) [30]. Since total inhibition of bacterial growth was achieved in all experiments at certain concentrations of the extract, it was possible to determine the MBC value. This is expressed as the mean value between the last concentration at which growth was detected and the first concentration at which no viable cells were present. Table 7 and Figure 4 show the values of IC50 and MBC obtained for each bacterium, expressed in % (*v*/*v*) of the extract in the reaction mixture.

## 3. Discussion

The development of multiresistant bacteria threatens the value of antibiotics, favoring the loss of their effectiveness. Currently, these microorganisms can spread between different countries and/or continents, making AMR a global problem. The low profitability for pharmaceutical companies involved in the development of new antibiotics makes it necessary to find other therapies that allow us to combat pathogenic bacteria, especially MDR ones. Plants and their secondary metabolites, including polyphenols, may be an alternative to alleviating the AMR problem.

According to the WHO classification, the bacteria tested in this study may fall into categories 1 and 2, with *P. aeruginosa* and *K. pneumoniae* belonging to category 1 (critical priority) due to their resistance to carbapenems, and *K. pneumoniae* because of its resistance to third-generation cephalosporins. For this last reason *P. mirabilis*, *E. coli*, *C. freundii*, *E. cloacae* and *S. enteritidis* can be included in category 1, with the latter also resistant to quinolones. *S. aureus* belongs to category 2 (high priority) due to its resistance to methicillin.

The white grape marc extract used in this study was shown to be rich in polyphenols and to have antioxidant [31,32] and antimicrobial [25,33] properties. The extraction process (medium-scale ambient temperature (MSAT) system) uses a generally recognized as safe (GRAS) solvent (ethyl lactate) [34] and complies with several of the principles of green analytical chemistry (GAC) [35] and with almost all the principles of green sample preparation (GSP) [36]: use of safe solvents, energy efficiency (as room temperature and atmospheric pressure are employed) and reuse of generated waste. This makes the process sustainable and safe. The MSAT technique patented by Lores et al. [22] was later used by Gato et al. [37] and again by Lores et al. [38] to extract polyphenols for multicomponent extracts from different varieties of blueberries (*Vaccinium corymbosum*) and from Scotch broom cuts (*Cytisus scoparius*), respectively, obtaining excellent results in both cases. A recent publication by the same research group employed this technique with GRAS solvents to optimize parameters for polyphenolic recovery from white grape marc extracts [39]. In this work, this system was chosen due to its high potential to extract bioactive compounds from natural sources and because it is a technique that complies with the principles of GAC and GSP. As a solvent, a 50% hydro–organic mixture based on ethyl lactate was used because of its higher extractive efficiency of polyphenols when compared with other solvents [40] and due to its reported efficacy when used for the extraction of polyphenols from *C. scoparius* [38], while at the same time being suitable for human consumption [41].

TPI was obtained using Otto Folin and Vintila Ciocalteu’s assay, which is the most commonly used method for this purpose. The TPI value of our grape marc extract was better than others found in the literature: Luchian et al. reported TPI results for white varieties of 2.76 and 2.03 mgGAE/mL in their study of the antimicrobial activity of grape marc extracts from different varieties of white and red grapes [42] compared with 8.27 mgGAE/mL in our study. Álvarez-Casas et al. gathered data for seven varieties of white grapes, obtaining between 22 and 44 mgGAE/g dry grape (compared with 43.4 mgGAE/g dry grape for the Albariño variety) [31]. However, they used a high-energy-consuming and difficult-to-scale method (PSE), which makes our methodology more appropriate, despite presenting a lower value of IPT (19.33 mgGAE/g dry grape marc).

Two different methods were used to determine the antioxidant activity of the extract, with both exhibiting similar results. It is common to use several techniques to determine the total antioxidant activity of a sample, which also facilitates the comparison with data from other authors. The main advantage of the ABTS method is the presence of absorption peaks at four different wavelengths, which reduces possible interferences [43]. Luchian et al. calculated the antioxidant activity using two techniques, including using DPPH [42], which is the same method used by Trošt et al. to evaluate this parameter [44], with both showing good results. Álvarez-Casas et al. also used this reagent, obtaining values of antioxidant activity for seven varieties of white grapes between 440.51 and 1068.75 mgTRE/g dry grape (compared with 913.56 mgTRE/g dry grape for the Albariño variety) [31]. These results are much higher than those obtained in our study (30.28 mgTRE/g dry grape marc), but as previously mentioned, they employed a technique that is less sustainable and scalable than MSAT. The ABTS reagent was used by González-Centeno et al. in grape marc extracts of four varieties of white grapes, obtaining results in the range of 71.6 to 134 mg TRE/g dry grape marc [45], which are higher values than those of this study (25.74 mg TRE/g dry grape marc). However, their extraction process was also based on PSE, with an additional previous freeze-drying stage, and therefore, requires a much larger energy consumption.

Polyphenols can be found in several foods present in our diet (fruits, vegetables, cereals, etc.), playing a very important role in human health. Their consumption is related to disease prevention [46] by modulating the intestinal microbiota: they allow for the growth of beneficial bacteria while eliminating pathogenic bacteria, which has to do with their antioxidant activity [47,48,49]. Their structure influences the antioxidant capacity, where hydroxyl groups are able to neutralize free radicals and other reactive oxygen species (ROS), and thus, their consumption prevents oxidative damage and reduces inflammation [50].

The most abundant polyphenols detected in our extract were coincident with the ones reported by other authors that evaluated the antibacterial activity of white grape marc extracts (Álvarez-Casas et al. [31], Rodríguez-Rama et al. [33], Pop et al. [51] and Trošt et al. [44]). In all cases, the predominant ones were flavonoids of the flavan-3-oles family, with catechin and epicatechin being the most abundant. There are, nevertheless, big differences in the nature of minority polyphenols. Different variables (part of the grape studied, variety, weather, geographical factors, vineyard, harvest year, production techniques), as well as polyphenolic extraction techniques and analytical detection methods used, may be responsible for this variation [19,30,46,52,53].

There is disagreement in the literature on whether the polyphenolic qualitative content of an extract is related to its antimicrobial activity, with some authors in favor [54,55] and others against [24,56]. The multiple mechanisms of action reported for polyphenols, possibly influenced by their chemical structure, may affect the cell wall and membrane, the formation of biofilms, the functionality of ion channels, bacterial metabolism, protein biosynthesis, inhibition of adenosine triphosphate (ATP) and nucleic acid synthesis, etc. [54,55,56,57,58].

Manso et al. [25] reviewed the activity of effective natural extracts against 7 of the 19 bacteria analyzed in this work. The most frequently studied was *S. aureus* (12 assays), followed by *E. coli* (6), *P. aeruginosa* (4), *K. pneumoniae* (4), *S. maltophilia* (3), *S. epidermidis* (2) and *E. faecalis* (1). We have not found any previous studies that evaluated the antibacterial activity of polyphenol-rich extracts against the rest of the bacteria tested in our study (*P. fluorescens, B. cepacia, P. mirabilis, C. freundii, E. cloacae, A. punctata, Y. enterocolitica, S. enteritidis, S. saprophyticus, S. agalactiae, S. pyogenes* or *E. faecium*). Seventeen of the articles reviewed by Manso et al. [25] evaluated species included in our study. Of these, 3 used exclusively pure polyphenols standards of synthetic origin and not natural extracts, 12 used only natural polyphenolic extracts, and 2 used both natural extracts and pure standards of the most abundant polyphenols present in the extracts used. These studies show that the interactions between polyphenols influence the activity of the extracts, with the antimicrobial activity of the mixture being higher than the one exhibited by each of them individually. These findings are supported by other studies and suggest that the antimicrobial activity of a mixture of polyphenols is a result of the synergistic action of these molecules [24,54,59].

We observed that bacteria belonging to related taxonomic groups present very different IC50 and MBC values, which makes it difficult to predict the effectiveness of the extract. This can be explained in terms of the complexity of our extract: aside from a very complex polyphenolic content, the composition of the extract includes other substances that can act synergistically or antagonistically with the active molecules. These substances might, in some cases, protect the bacterial cells from damage, or in other cases make them more sensitive to the action of polyphenols. As a wide variety of mechanisms of action have been reported for different polyphenols, slight variations in the metabolism of the bacterial strains can result in behavior changes when subjected to the action of such complex extracts.

Combining antibiotics with polyphenols or polyphenolic extracts against clinical strains is also interesting for determining the potential use of natural extracts as an alternative or complement to antibacterial therapy. Several studies evaluated this synergism against clinical strains and, although some of them did not show this effect [58,60], others reported promising results [61,62,63].

Despite the known antimicrobial potential of polyphenolic extracts from grape residues against bacteria [19,64,65], fungi [57,66] and viruses [67], not many studies report the use of white grape extracts against bacteria from clinical samples. Three out of the four studies found used species coinciding with those of our clinical isolates (*S. aureus, E. coli, E. faecalis* and *K. pneumoniae*) [45,68,69].

In summary, on the basis of the in vitro experiments carried out in this work, it was shown that the extract used has antibacterial activity against all the clinical strains studied, which makes it a potential alternative as an antibacterial therapy. Synergy studies with antibiotics would evaluate the improvement of their effectiveness and the possibility of reducing the antibiotic dose, thereby diminishing side effects. In vivo studies are needed to determine the real potential of this extract as a treatment for human infections. Further investigation into the mechanisms of action of polyphenols would also be useful. Future research should focus on these three aspects.

## 4. Materials and Methods

### 4.1. Clinical Strains

Nineteen bacteria were selected from clinical samples of patients. These samples were initially processed in the microbiology laboratories of the previously mentioned Galician hospitals. The choice of strains was based on their clinical relevance as bacterial pathogens in human infections, as well as their levels of antibiotic resistance. Twelve strains of Gram-negative bacilli were evaluated, 10 of which presented a multidrug-resistance profile, and an acquired resistance mechanism was detected in 6 of them. The remaining strains corresponded to Gram-positive cocci, in two of which an acquired resistance mechanism was detected (*S. aureus* and *S. epidermidis* are resistant to methicillin). After the selection of the strains, they were subcultured and stored in 1.5 mL tubes at −80 °C in a mixture of glycerol:distilled water (50:50) until further processing in the microbiology laboratories of the Faculty of Pharmacy of the University of Santiago de Compostela. The strains were grown in soy agar tripticase (TSA) for the experiments.

Fully automated systems were used for the identification and antibiotic susceptibility of these isolates. In HPM, combined panels of the MicroScan WalkAway system (Beckman Coulter, Brea, CA, United States) were used to perform identification at the species level and antibiogram. At the CHUF, matrix-assisted laser desorption/ionization time of flight (MALDI-TOF, Biomerieux, France) was used for bacterial identification and the Vitek-2 system (Biomerieux, France) for their antibiogram. MicroScan and Vitek-2 use the broth microdilution technique to perform antibiotic susceptibility studies. Briefly, a standardized suspension of the bacteria was obtained and inoculated into the MicroScan or Vitek-2 panel, which was introduced into the equipment and incubated at 37 °C. The determination of susceptibility or resistance to the antibiotics tested was measured using the minimum inhibitory concentration (MIC) based on the cut-off points established by EUCAST. When the detected resistance was absent in the natural phenotype, appropriate tests were carried out using molecular or phenotypic methods in order to identify the resistance mechanism. Thus, carbapenemase enzymes were detected using PCR (GeneXpert, Cepheid, Sunnyvale, CA, USA) and ESBL enzyme and methicillin resistances were detected using antibiotic discs.

### 4.2. Extract

In this work, we employed an extract that came from Galician white grape marc (*Vitis vinifera,* var. Albariño) obtained from the harvest of 2021. The grapes were grown in the Rías Baixas Designation of Origin subzone O Salnés in the *Mar de Frades* winery (San Martiño de Meis, Pontevedra, Spain). Initially, the grapes were pressed in the winery using the usual procedure for obtaining the must, and the grape marc was immediately collected and frozen at −21 °C in food-grade plastic bags (20 × 20 cm) that were hermetically sealed to reduce possible degradation.

### 4.3. Obtaining the Extract

The technique used in this work to obtain the extract was extraction by means of a medium-scale ambient temperature (MSAT) system, which was developed and patented by Lores et al. for the recovery of polyphenols from white grape marc samples at the laboratory and pilot scales [22]. Briefly, the grape marc was weighed and crushed using a porcelain mortar, where it was subsequently mixed with washed sea sand (average diameter 0.25 mm, Scharlau, Barcelona, Spain), which eases the breaking of the grape marc and the release of the bioactive compounds it contains. This homogeneous mixture was introduced sequentially into the extraction column (20 cm high glass cylinder with an external diameter of 5 cm with a 160–250 μm pore filter plate) and conical bottom with a wrench to regulate the flow), through which the extracting solvent (50% ethyl lactate–water) was passed on. The process took 40 min.

### 4.4. Total Polyphenolic Index of the Extract (TPI)

To estimate the polyphenolic index (TPI) of the extract, the Folin–Ciocalteu method was used, adapting the protocol proposed by Singleton et al. [70] for microtitration in 96-well plates. A total of 20 μL of the diluted extract was mixed with 100 μL of Folin–Ciocalteu reagent (1:10) (Sigma-Aldrich Gm bH, Steinheim, Germany) and 80 μL of a sodium carbonate solution (7.5 g/L) (Panreac, Barcelona, Spain). The mixture was homogenized and, after 30 min in the dark, its absorbance was measured at 760 nm. For this purpose, the SPECTROstar Nano UV/Vis microplate reader, 200–1000 nm (BMG Labtech, Ortenberg, Germany), was used. The final volume of each well of the 96-well microplates used was 200 μL. The TPI was calculated from a calibration line obtained via reacting the Folin reagent with different concentrations of gallic acid (Sigma Aldrich GmbH, Steinheim, Germany) in a range between 20 and 160 mg/L (0.200–0.800 absorbance units (UA)). All the experiments were performed in triplicate. This calibration line was obtained by representing the equivalent gallic acid concentration (GAE) against the measured signal (absorbance). The equation of the calibration line was obtained using linear least squares regression and the coefficient of determination (R^2^) indicated the quality of the fit. The TPI results were expressed as milligrams of gallic acid equivalent per liter of extract (mgGAE/L).

### 4.5. Antioxidant Activity (AA) of the Extract

The radical source DPPH (Sigma, St. Louis, MO, USA) and ABTS (Glentham Life Sciences, Corsham, UK) was used to calculate the antioxidant capacity of the extract and its mean maximum inhibitory concentration IC50.

The method described by Zhang et al. using the DPPH reagent [71] was used, with slight modifications. Ninety-six-well plates were employed, in which 100 μL of each serial dilution of the extract and 100 μL of the DPPH solution (0.25 mM in methanol) were mixed. Ultrapure water MilliQ, which was produced in the laboratory with a Milli-Q gradient system (Millipore, Bedford, MA, USA), was used as a control target. The plate was shaken for 10 s, keeping it in the dark for 10 min. Absorbance was measured at 515 nm by employing the previously used microplate reader (SPECTROstar Nano).

The procedure described by Xiao et al. [72] was employed with slight modifications in the ABTS assay. Initially, to generate the ABTS+ cation, a stock solution (7 mM ABTS in ethanol) was used and reacted with 2.45 mM potassium persulfate (Glentham Life Sciences) dissolved in ethanol. The mixture was kept in the dark at 25 °C for 16 h and dissolved in ethanol to obtain an absorbance of 0.700 at 752 nm. Serial dilutions of the extract were performed in a 96-well microplate and 50 μL of each concentration were mixed with 200 μL of the previous mixture (in which we added ABTS+), shaken and stored in the dark for 7 min. A color change occurred as the cation was reduced by the polyphenols of the extract. In this case, absorbance was measured at 752 nm (using the SPECTROstar Nano microplate reader). The determinations were made in triplicate.

In order to calculate the mean inhibitory index IC50 using both methods, a representation of the change in absorbance versus the concentrations of the sample was made. Two points with a 50% inhibition ratio were selected and a regression line was drawn. The sample concentration was calculated by replacing the absorbance value with 50% in the regression equation obtained.

Inhibition of DPPH and ABTS radicals was quantified in the same way as with the extract but instead using 6-hydroxy-2,5,7,8-tetramethylchromo-2-carboxylic acid (Trolox, Sigma-Aldrich Gm bH, a synthetic analog of vitamin E), which was used as a standard, over a concentration range of 12 to 124 μM (0.200–0.800 AU) when DPPH was used and 32 to 160 μM (0.200–0.700 AU) with ABTS. Triplicate assays were conducted. Antioxidant activity was expressed as millimoles equivalent of Trolox per liter of extract (mmolTRE/L).

### 4.6. Characterization of Polyphenols Using Liquid Chromatography Coupled to a Tandem Mass Spectrometer (LC-MS/MS)

LC-MS/MS was used to determine and quantify the polyphenols in the extract. This study was performed on a TSQ Quantum UltraTM triple quadrupole mass spectrometer equipped with a heated electrospray ionization source (HESI-II) (Thermo Scientific, San Jose, CA, USA) and an Accela Open self-sampler with a 20 μL loop. Instrumental conditions previously optimized by Skoko et al. were used [73]. For the chromatographic separation, the column employed was Kinetex C18 (2.6 μm, 100 × 2.1 mm), which was obtained from Phenomenex (Torrance, CA, USA) and set at 50 °C. Water (A) (ultrapure water Milli-Q) and methanol (B) (Scharlau, Barcelona, Spain) were used as the mobile phase, both with 0.1% formic acid (98–100%, Merck, Darmstadt, Germany). Elution was performed in a gradient, initially using 5% B for 5 min. This increased to 90% B at 11 min and remained constant for 3 min. Finally, the initial conditions were reached at 5 min. The injection volume was 10 μL and the flow rate of the mobile phase was 200 μL/min. Each injection was performed at 20 min. An individual direct infusion was used to optimize the MS/MS parameters for all polyphenols under study, and the most abundant collision-induced fragments were considered for quantification. Other parameters of the HESI source were a spray voltage of 3000 V, vaporizer temperature of 350 °C, 35 arbitrary units (au) of ambient gas pressure, 0 au of ion-sweeping pressure, 10 au of auxiliary gas pressure and a capillary temperature of 320 °C.

Pure polyphenol standards were employed to correctly identify those contained in the extract and are shown in Table 8. With them, stock standard solutions (between 1000 and 3000 mg/L, depending on the compound) were prepared using methanol as the solvent for all polyphenols, except for gallic acid, for which methanol/water was employed. Working solutions and calibration standards were prepared from these stock solutions. The standards were diluted with Milli-Q water as the solvent. For the calibration standards, a mixture of water–methanol at 50% was used. These standards were injected in SRM (selected reaction monitoring) mode, that is, selecting and optimizing only the most specific fragments of each compound. The extract was then injected. Quantification was performed using the first mass transition, while the second and the next (when available) were employed for identification and/or confirmation. In the case of having only two transitions, the polyphenolic identity was confirmed using these transitions, as well as retention times. Polyphenols were detected by working simultaneously in negative and positive modes, choosing for each individual polyphenol the mode with which the best results were obtained. The standards of the target polyphenols were introduced into the mass spectrometer via flow injection, and the collision energies of the SRM transitions were optimized for each polyphenol. The management of the instrument and data processing was carried out using the Xcalibur 2.2 software package (Thermo Fisher Scientific, Waltham, MA, USA).

Thirteen outstanding polyphenols, shown in Table 9, were detected. In addition, retention times (Rt, in minutes), molecular mass (in g/mol), ionization mode, MS/MS transitions, linear ranges and coefficients of determination (R^2^) for the identification of each polyphenol are presented.

To quantify the extract phenolics, calibration lines for each identified polyphenol were performed. The concentration range used was between 0.5 and 5 mg/L. Procyanidins B1, B2 and C1 were quantified together as equivalent total procyanidin B1, employing the Trace FinderTM 3.2 software (Thermo Scientific).

### 4.7. Determination of Antibacterial Activity from the Viable Cell Count Method with Fluorometric Reading

Antibacterial activity was determined using the viable cell count method with fluorometric reading. For this purpose, 100 μL of a bacterial concentration of 10^6^ colony forming units (CFU)/mL in Cation Adjusted Müller Hinton II broth (CAMBH) purchased from Becton-Dickinson (BBL, Sparks, NV, USA) was mixed with 100 μL of each of the extract concentrations used (0%, 0.625%, 1.25%, 2.5%, 5%, 10% and 20%) in a 96-well microplate and incubated at 37 °C for 16 h. A blank of the extract was employed by incubating 100 μL of CAMBH broth instead of the bacterial culture. After the incubation, 100 μL of fresh CAMBH, 60 μL of saline phosphate buffer (PBS, 1M), 20 μL of resazurin (ThermoFisher Scientific, Massachusetts, USA) and 20 μL of the corresponding bacterial inoculum were mixed in another microplate with 96 wells and incubated at 37 °C. Fluorometric reading was performed to determine the number of viable cells present in each well at an excitation wavelength of 544 nm and emission wavelength of 590 nm using the FLUOstar microplate reader with a spectral range of 240–740 nm (BMG Labtech). The experiments were performed in triplicate for the bacteria and in duplicate for the target. Calibration lines were carried out for each bacterium. Each experiment was performed three times independently, considering the test result valid when the order of magnitude of the number of viable cells was the same between each replica. Otherwise, the experiment was discarded and repeated.

For the fluorometric reading, the alamarBlue system was used, which employs a weakly fluorescent indicator, namely, resazurin, which is reduced to fluorescent resorufin by metabolically active bacteria, causing a discoloration [74]. Its use makes it easier to read results when using the broth microdilution method [75], as shown by several studies [71,76,77,78]. Since the incubation duration and number of cells seeded greatly affect the values obtained, it is very important to optimize these two variables for good reproducibility and robust data [79]. For this reason, the experiments performed attempted to obtain the same initial bacterial concentration in each assay and they were considered valid only if the number of viable cells was of the same order of magnitude between each replicate.

## 5. Conclusions

The 50% hydro–organic mixture of ethyl lactate has proven to be an effective solvent for the extraction of polyphenols from Albariño white grape marc, which allows to obtain a high total polyphenol index (TPI). The fact of using the MSAT system, together with the categorization of ethyl lactate as a GRAS solvent, makes this process environmentally friendly. The method of detecting viable cells with fluorometric reading has been considered a good technique to determine the antibacterial activity of the extract used since the results have been reproducible, and the technique is robust, accurate and rapid. Albariño white grape marc extract was shown to be effective against all the assayed clinical strains, as low IC50 and MBC values were obtained for all of them. This effectiveness was independent of the antibiotic resistance level of the different bacteria tested, which makes the extract a potential alternative in the fight against AMR. The different pattern of each bacterium toward the extract, even for phylogenetically close ones, was possibly due to the variety of mechanisms of action of polyphenols, their synergy and the complexity of the extract, which makes the results unpredictable. The experiments performed were preliminary studies that could be the basis of future investigations on the effectiveness of this extract to combat real infections caused by multidrug-resistant bacteria. More research is needed in this field to make polyphenols a useful tool as an antibacterial therapy. In order to verify the true potential of the extract for the treatment of infections, it would also be necessary to carry out in vivo studies to confirm the results obtained in vitro.

## Figures and Tables

**Figure 1 pharmaceuticals-16-00950-f001:**
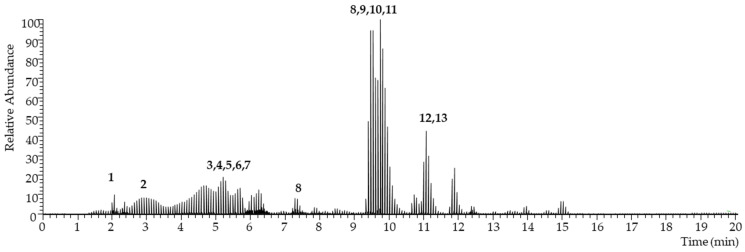
Total ion chromatogram of the grape marc extract. See the code correspondence in Figure 2.

**Figure 2 pharmaceuticals-16-00950-f002:**
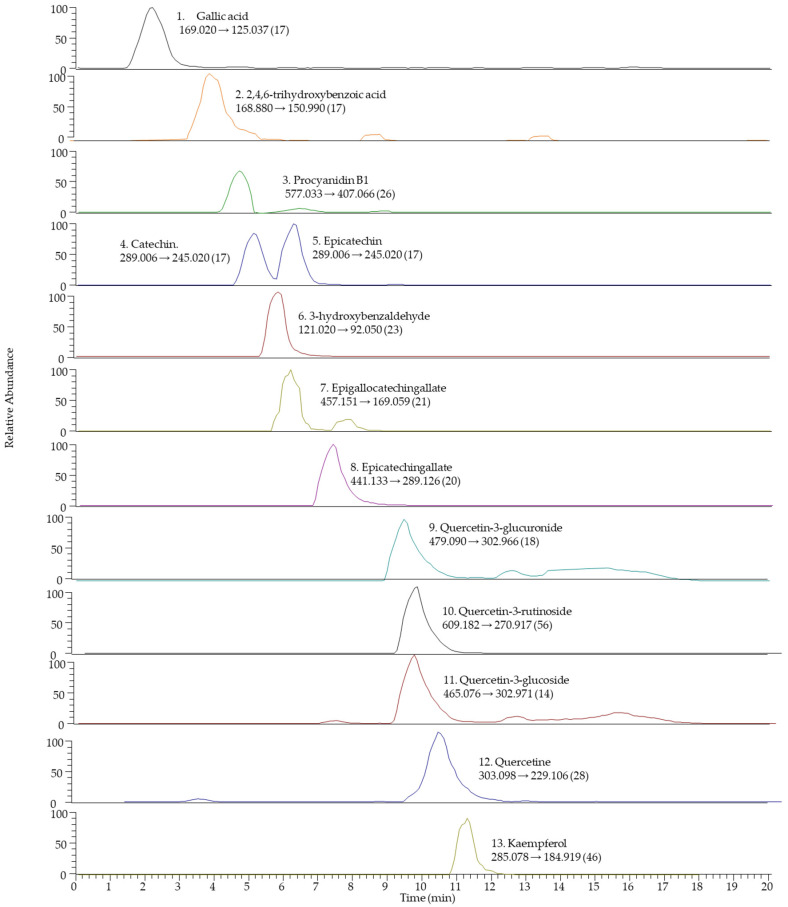
SRM-reconstructed chromatograms for the 13 target polyphenols detected in the white grape marc extract.

**Figure 3 pharmaceuticals-16-00950-f003:**
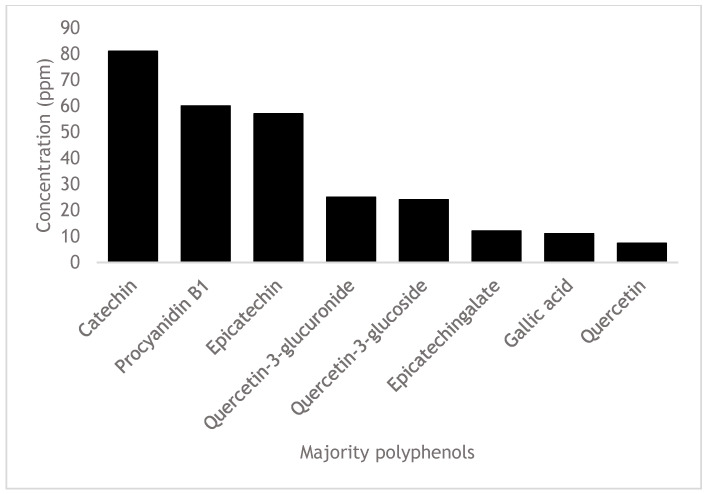
Detected polyphenols in the Albariño white grape marc extract and their concentration in ppm (*m/v*), categorized as majority and minority fractions.

**Figure 4 pharmaceuticals-16-00950-f004:**
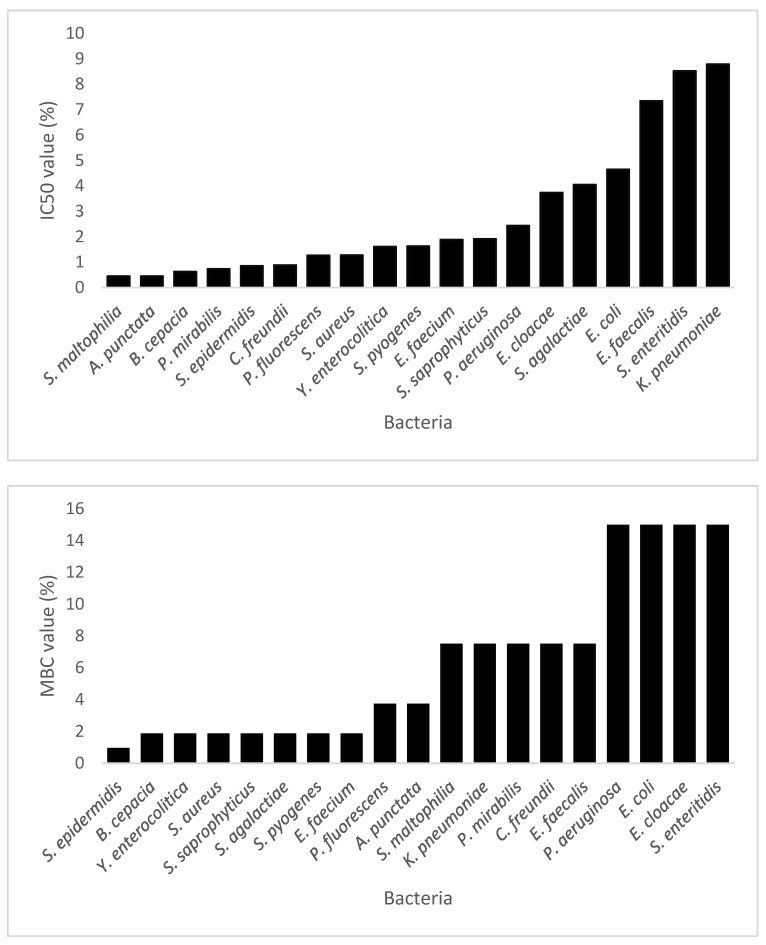
Sensitivity of the studied bacteria to Albariño white grape marc extract expressed as IC50 and MBC values in % (*v*/*v*). Average values for triplicates were used to calculate IC50 concentrations using the Quest Graph IC50 Calculator (AAT Bioquest 2022).

**Table 1 pharmaceuticals-16-00950-t001:** Classification of the main polyphenols according to the number of carbon atoms: classes, subclasses and representatives of the main polyphenols found in white grapes.

Number and Layout of Carbon Atoms	Polyphenol Class or Family (Polyphenol Subclass)	Main Representatives in White Grapes
C_6_-C_1_	Benzoic acids and derivatives	Gallic acid 2,4,6-trihydroxibenzoic acid 3-hydrobenzaldehyde Protocatechuic acid Syringic acid
C_6_-C_3_	Cinnamic acids	Caftaric acid Caffeic acid
	Coumarins	
C_6_-C_2_-C_6_	Stilbens	
C_6_-C_3_-C_6_ Flavonoids	*Flavonols*	Quercetin-3-glucuronide (miquelianin) Quercetin-3-O-rutinoside (rutin) Quercetin-3-O-glucoside (isoquercitrin) Quercetin Kaempferol Quercetin-3-rhamnoside (quercitrin)
	*Flavan-3-ols*	Procyanidins Catechin Epicatechin Epigallocatequingalate Epicatequingalate
	*Flavanones*	
	*Flavones*	Luteolin
	*Isoflavones*	
	*Anthocyani(di)ns*	
(C_6_-C_1_)_n_	Hydrolyzable tannins	
	*Ellagitannins*	
	*Gallotanins*	

**Table 2 pharmaceuticals-16-00950-t002:** Clinical strains of Gram-negative bacilli obtained from HPM and CHUF.

Strain Code	Bacteria	Acquired Resistance Mechanism	Clinical Sample	Culture Medium for Their Isolation
**Pflu**	*Pseudomonas fluorescens*	VIM carbapenemase	Urine	CLED
**Paer**	*Pseudomonas aeruginosa*		Wound	BA
**Bcep**	*Burkholderia cepacia*		Catheter	BA
**Smal**	*Stenotrophomonas. maltophilia*		Wound	BA
**Kpne**	*Klebsiella pneumoniae*	OXA-48 carbapenemase	Urine	CLED
**Pmir**	*Proteus mirabilis*	ESBL	Urine	CLED
**Ecol**	*Escherichia coli*	ESBL	Urine	CLED
**Cfre**	*Citrobacter freundii*	ESBL	Urine	CLED
**Eclo**	*Enterobacter cloacae*		Urine	CLED
**Apun**	*Aeromonas punctata (caviae)*		Feces	CIN
**Yent**	*Yersinia enterocolitica*		Feces	CIN
**Sent**	*Salmonella enteritidis*	ESBL	Feces	SS

**Table 3 pharmaceuticals-16-00950-t003:** Clinical strains of Gram-positive cocci obtained from HPM.

Strain Code	Bacteria	Acquired Resistance Mechanism	Clinical Sample	Culture Medium for Their Isolation
**Saur**	*Staphylococcus aureus*	Methicillin resistant	Blood culture	BA
**Sepi**	*Staphylococcus epidermidis*	Methicillin resistant	Wound	BA
**Ssap**	*Staphylococcus saprophyticus*		Urine	CLED
**Saga**	*Streptococcus agalactiae*		Blood culture	BA
**Spyo**	*Streptococcus pyogenes*		Blood culture	BA
**Efae**	*Enterococcus faecalis*		Urine	CLED
**Efac**	*Enterococcus faecium*		Blood culture	BA

CLED: cysteine lactose electrolyte deficient agar, BA: sheep blood agar, CIN: cefsulodin-irgasan-novobiocin agar, SS: Salmonella Shigella agar, ESBL: extended-spectrum beta-lactamase-producing enzyme.

**Table 4 pharmaceuticals-16-00950-t004:** Antibiotics tested against the Gram-negative bacilli used and interpretation of the tests according to EUCAST guidance (CLSI for BCEP).

	AK	AM	AS	AUG	AZT	C/T	CZA	CAZ	CFE	CFT	CFX	CL	CP	CPE	CRM	ETR	FD	FOS	GM	IMP	LVX	MER	NXN	PI	PT	TS	TI	TO
Pflu	S				R	R	R	R				S	R	R	R				R	R	R	R			R		R	R
Paer	S		R		S			R		R		S	R	R					S	R	R	R		R	R	R	R	R
Bcep								R													R	S				S		
Smal	R				R			R		R		R	R	R					R	R	S			R	R	S	R	R
Kpne		R	R	R		R	S	R	R	R	R	S	R	R	R	R	R	R	R	R	R		R		R	R	R	R
Pmir	S	R		S				R	R	R	S	R	R	R	R	S	R	R	S	I	S		R		S	S	R	S
Ecol	S	R		S				R	R	R	S	S	R	R	R	S	S	S	S	S	R		R		S	R	R	S
Cfre	R	R		R				R	R	R	R	S	R	R	R	S	S	S	R	S	R		R		R	R	R	R
Eclo	S	R		R	R	R	S	R	R	R	R	S	R	R	R	R	R	S	R	S	R	S	R	R		R	R	R
Apun	S							S		S *			S	S	S				S			S			S	S		
Yent	S	R		R				S		S *			R		S	S		S	S			S				S		
Sent	R	R		R				R		R *			R	R	R	S		S	R			S			S	R		

* Tested ceftriaxone. AK: amikacin, AM: ampicillin, AS: ampicillin-sulbactam, AUG: amoxicillin-clavulanic acid, AZT: aztreonam, C/T: ceftolozane-tazobactam, CZA: ceftazidime-avibactam, CAZ: ceftazidime, CFE: cefixime, CFT: cefotaxime, CFX: cefoxitin, COL: colistin, CP: ciprofloxacin, CPE: cefepime, CRM: cefuroxime, ETR: ertapenem, FD: nitrofurantoin, FOS: phosphomycin, GM: gentamicin, IMP: imipenem, LVX: levofloxacin, MER: meropenem, NXN: norfloxacin, PI: piperacillin, PT: piperacillin-tazobactam, TS: cotrimoxazole, TI: ticarcillin, TO: tobramycin.

**Table 5 pharmaceuticals-16-00950-t005:** Antibiotics tested against the Gram-positive cocci used and interpretation of the tests according to EUCAST guidance.

	AK	AM	AUG	CD	CP	DAP	E	ES1000	FA	FD	FOS	GM	GM500	LVX	LZD	MUP	OX	P	RIF	SYN	TS	TE	TEI	TO	VA
Saur	S		R	S	R	S	S		S		S	S		R	S	S	R	R	S		S	S	S	S	S
Sepi	S		R	S	R	S	S		S		S	R		R	S	R	R	R	S		R	I	S	R	S
Ssap		R	S	S	S	S	R			S	R	S		S	S		S	R			S	S	S		S
Saga		S	S	S	S		S					R		S				S			R				S
Spyo				S			S								S			S							S
Efae		S		R	R	S	R	S		S		R	S	R	S			S		R	R		S		S
Efac		R		R	R	S	R	R					S	R	S			R		s	S		S		S

AK: amikacin, AM: ampicillin, AUG: amoxicillin-clavulanic acid, CD: clindamycin, CP: ciprofloxacin, DAP: daptomycin, E: erythromycin, ES1000: streptomycin 1000, FA: fusidic acid, FD: nitrofurantoin, FOS: phosphomycin, GM: gentamicin, GM500: gentamicin 500, LVX: levofloxacin, LZD: linezolid, MUP: mupirocin, OX: oxacillin, P: penicillin, RIF: rifampicin, SYN: synercid, TS: cotrimoxazole, TE: tetracycline, TEI: teicoplanin, TO: tobramycin, VA: vancomycin.

**Table 6 pharmaceuticals-16-00950-t006:** Quantification in ppm (*m/v*) of the polyphenols detected in the grape marc extract.

Polyphenol	Concentration (ppm)
Gallic acid	11 ± 1
2,4,6-trihydroxybenzoic acid	1.2 ± 0.08
Procyanidin B1	60 ± 5
Catechin	81 ± 7
3-hydroxibenzoaldehide	0.04 ± 0.007
Epicatechin	57 ± 4
Epigallocatequingalate	0.1 ± 0.008
Epicatechingalate	12 ± 2
Quercetin-3-glucuronide	25 ± 5
Quercetin-3-rutinoside	0.5 ± 0.17
Quercetin-3-glucoside	24 ± 3
Quercetin	7.3 ± 0.9
Kaempferol	1.5 ± 0.2

**Table 7 pharmaceuticals-16-00950-t007:** Values of IC50 and MBC of clinical strains, expressed in % (*v*/*v*).

Strain Code	IC50 (% (*v*/*v*))	MBC (% (*v*/*v*))
Pflu	1.27	3.75
Paer	2.44	15
Bcep	0.63	1.87
Smal	0.45	7.5
Kpne	8.79	7.5
Pmir	0.74	7.5
Ecol	4.65	15
Cfre	0.89	7.5
Eclo	3.74	15
Apun	0.45	3.75
Yent	1.61	1.87
Sent	8.52	15
Saur	1.28	1.87
Sepi	0.86	0.97
Ssap	1.92	1.87
Saga	4.05	1.87
Spyo	1.63	1.87
Efae	7.34	7.5
Efac	1.89	1.87

**Table 8 pharmaceuticals-16-00950-t008:** Characteristics of the polyphenols’ standards employed.

Polyphenols	Purity	Company	CAS
Gallic acid	99.9	SIGMA ^a^	149-91-7
2,4,6-trihydroxybenzoic acid	98.4	SIGMA ^a^	487-70-7
Procyanidin B1	96.7	EXTRAS ^b^	20315-25-7
Catechin	98.0	SIGMA ^a^	18829-70-4
3-hydroxybenzaldehyde	99.0	SIGMA ^a^	90-02-8
Epicatechin	90.0	SIGMA ^a^	490-46-0
Epigallocatechingalate	99.1	SIGMA ^a^	989-51-5
Epicatechingallate	98.0	SIGMA ^a^	1257-08-5
Quercetin-3-glucuronide	98.5	SIGMA ^a^	27253-19-6
Quercetin-3-rutinoside	99.1	SIGMA ^a^	115888-40-9
Quercetin-3-glucoside	98.0	SIGMA ^a^	21637-25-2
Quercetin	96.0	SIGMA ^a^	117-39-5
Kaempferol	99.3	SIGMA ^a^	520-18-3

^a^ Sigma Aldrich GmbH (Steinheim, Germany), ^b^ Extrasynthese (Genay, France).

**Table 9 pharmaceuticals-16-00950-t009:** Conditions of each detected polyphenol in the white grape marc extract.

Polyphenols	Rt (min)	Molecular Mass (g/mol)	I ^a^	Precursor Ion (m/z) ^b^	Product Ions (m/z) ^b^	Collision Energy (eV) ^b^	Linear Range (mg/L)	R^2^
**Gallic acid**	2.35	170.12	-	169.020	125.037 153.10	17 15	0.5–5	0.9972
**2,4,6-trihydroxybenzoic acid**	3.88	170.11	-	168.88	150.99 83.02 107.02	17 23 22	0.5–5	0.9960
**Procyanidin B1**	5.30	578.52	-	577.033	407.066 288.93 424.98	26 25 26	0.5–5	0.9960
**Catechin**	5.34	290.27	-	289.006	245.020 203.12	17 22	0.5–5	0.9995
**3-hydroxy-benzaldehyde**	5.77	122.12	-	121.02	92.05 93.05 120.04	23 20 19	0.5–5	0.9954
**Epicatechin**	6.50	290.27	-	289.006	245.020 203.12	17 22	0.5–5	0.9980
**Epigallocatechingalate**	6.80	458.4	-	457.151	169.059 125.09 305.09	21 42 21	0.5–5	0.9902
**Epicatechingalate**	7.29	442.4	-	441.133	289.126 125.08 169.05	20 42 24	0.5–5	0.9901
**Quercetin-3-glucuronide**	9.54	478.36	+	479.090	302.966 461.50	18 14	0.5–5	0.9998
**Quercetin-3-rutinoside**	9.72	610.518	-	609.182	270.917 178.87	56 44	0.5–5	0.9987
**Quercetin-3-glucoside**	9.75	464.376	+	465.076	302.971 256.90	14 41	0.5–5	0.9920
**Quercetin**	10.72	302.23	+	303.098	229.106 153.05	28 33	0.5–5	0.9976
**Kaempferol**	11.89	286.24	-	285.078	184.919 239.13	46 35	0.5–5	0.9957

^a^ I: ionization, ^b^ transitions underlined: employed for the quantification.

## Data Availability

Data is contained within the article.

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
