# Peer review of "Antibacterial Activity against Clinical Strains of a Natural Polyphenolic Extract from Albariño White Grape Marc"

_pharmaceuticals, 2023, doi:10.3390/ph16070950_

Round 1

Reviewer 1 Report

Dear Editor/Author;

-          Abstract should be detailed.

-          The introduction should be expanded with the areas of use of natural antibacterial agents. For this, it is recommended to use the following articles.

Coloring of Woolen Fabrics with Natural Resources and Investigating the Color Perceptions of Children on These Fabrics

Use of tea and tobacco industrial wastes in dyeing and antibacterial finishing of cotton fabrics

Natural functionalisation of a traditional textile "Ehram"

Investigation of Antibacterial Effect of Hypericum perforatum L. on Woolen Fabrics

Application of Glycyrrhiza glabra L. Root as a Natural Antibacterial Agent in Finishing of Textile

Investigating the usage of eucalyptus leaves in antibacterial finishing of textiles against Gram-positive and Gram-negative bacteria

Treatment of originally coloured wools with garlic stem extracts and zinc chloride to ensure anti-bacterial properties with limited colour changes

Use of Alpinia Officinarum Rhizome in Textile Dyeing and Gaining Simultaneous Antibacterial Properties

-          Emphasis should be placed on the purpose of the article.

-          The article should be reviewed by a native English reader.

-          Grammatical errors should be corrected throughout the article.

-          The conclusion should be strengthened.

-          The contribution of this article to future studies should be stated.

-          It should be clearly stated in which area this article fills the gap.

Dear Editor/Author;

-          Abstract should be detailed.

-          The introduction should be expanded with the areas of use of natural antibacterial agents. For this, it is recommended to use the following articles.

Coloring of Woolen Fabrics with Natural Resources and Investigating the Color Perceptions of Children on These Fabrics

Use of tea and tobacco industrial wastes in dyeing and antibacterial finishing of cotton fabrics

Natural functionalisation of a traditional textile "Ehram"

Investigation of Antibacterial Effect of Hypericum perforatum L. on Woolen Fabrics

Application of Glycyrrhiza glabra L. Root as a Natural Antibacterial Agent in Finishing of Textile

Investigating the usage of eucalyptus leaves in antibacterial finishing of textiles against Gram-positive and Gram-negative bacteria

Treatment of originally coloured wools with garlic stem extracts and zinc chloride to ensure anti-bacterial properties with limited colour changes

Use of Alpinia Officinarum Rhizome in Textile Dyeing and Gaining Simultaneous Antibacterial Properties

-          Emphasis should be placed on the purpose of the article.

-          The article should be reviewed by a native English reader.

-          Grammatical errors should be corrected throughout the article.

-          The conclusion should be strengthened.

-          The contribution of this article to future studies should be stated.

-          It should be clearly stated in which area this article fills the gap.

Author Response

Reviewer 1 thinks the following aspects of the manuscript should be improved:

- Abstract should be detailed.

The abstract has been reviewed, and hopefully improved.

      - The introduction should be expanded with the areas of use of natural antibacterial agents. For this, it is recommended to use the following articles.

Coloring of Woolen Fabrics with Natural Resources and Investigating the Color Perceptions of Children on These Fabrics

Use of tea and tobacco industrial wastes in dyeing and antibacterial finishing of cotton fabrics

Natural functionalisation of a traditional textile "Ehram"

Investigation of Antibacterial Effect of Hypericum perforatum L. on Woolen Fabrics

Application of Glycyrrhiza glabra L. Root as a Natural Antibacterial Agent in Finishing of Textile

Investigating the usage of eucalyptus leaves in antibacterial finishing of textiles against Gram-positive and Gram-negative bacteria

Treatment of originally coloured wools with garlic stem extracts and zinc chloride to ensure anti-bacterial properties with limited colour changes

Use of Alpinia Officinarum Rhizome in Textile Dyeing and Gaining Simultaneous Antibacterial Properties

Thank you very much for the suggestion. However, we believe that none of the publications suggested by Reviewer 1 contribute relevant information to our study, since we have already provided sufficient bibliography on the subject.

-          Emphasis should be placed on the purpose of the article.

We appreciate the suggestion, but the main purpose of our study is clearly stated in the last paragraph of the introduction: there are very few studies which evaluate the antimicrobial activity of polyphenol-rich extracts on clinical isolates.

-          The article should be reviewed by a native English reader.

 Comprehensive editing of English has been done.

-          Grammatical errors should be corrected throughout the article.

 The grammatical errors detected have been corrected.

-          The conclusion should be strengthened.

 The conclusion has been rewritten in order to improve it.

-          The contribution of this article to future studies should be stated.

 A sentence has been added to the conclusion to clarify it.

-          It should be clearly stated in which area this article fills the gap.

      As stated above, there is a lack of bibliography on the antibacterial activity of polyphenol-rich extracts against clinical isolates. Our article reports the antibacterial activity of such an extract against the largest collection -to our knowledge- of clinical isolates analyzed to date.

Reviewer 2 Report

The topic is interesting but needs more experimentation to substantiate the claim by the authors:

1. TFC need to be done

2. Qualitative phytochemical analysis needs to be done.

3. For antioxidant activities some more experimentation needs to be done e.g. FRAP, H2O2, CUPRAC etc.

4. FTIR need to be done.

5. MIC experimentation needs to be done. Figures are essential to prove the antibacterial effect. 

Minor Comments:

1. Reference style needs to be corrected.

2. Figure quality should be improved. 

Minor editing of the English language required

Author Response

Reviewer 2 finds the topic interesting but thinks that more experimentation should be done.

  1. TFC need to be done

The polyphenolic content has been expressed in terms of TPI (Total Polyphenolic index), which includes both flavonoid content (TFC) and non-flavonoid polyphenols. Both are, in any case, estimated indexes. However, detailed analysis by LC-MS/MS allows not only to identify the main flavonoid and non-flavonoid polyphenols, but also to quantify them. This information, detailed in Table 7, characterizes in detail the polyphenolic profile of the extract.

  1. Qualitative phytochemical analysis needs to be done.

The main components of the extract -and the ones relevant for these study- are polyphenols. Identification and quantification of the main polyphenols have already been done and reported in table 7, as we have explained above.

  1. For antioxidant activities some more experimentation needs to be done e.g. FRAP, H2O2, CUPRAC etc.

Two relevant techniques have already been used. DPPH and ABTS radicals are between the most relevant radicals to detect antioxidant activity. The main objective of the experiments is the evaluation of the antibacterial activity of the extract against clinical samples, although antioxidant activity is an important complementary parameter to describe in depth the nature of the extract. We think that two important techniques are enough to show the high antioxidant capacity of the extract. Furthermore, we remark the great correlation between the results obtained with both techniques.

  1. FTIR need to be done.

FTIR measures the absorption of infrared radiation by a sample and provides information on the functional groups present in the sample. Its main objective is the qualitative characterization of a sample for identification purposes, ideally for pure and/or not very complex substances, highlighting its application to the identification of polymers.

Our extract is a complex mixture, so identification with this type of analytical tool is of little use. For this reason, the identification of the main bioactive compounds is carried out by LC-MS/MS. The first dimension of this technique, liquid chromatography (LC), allows the separation of the compounds prior to their identification. The second dimension, tandem mass spectrometry (MS/MS), allows the unequivocal identification (together with the chromatographic retention time as an additional parameter) of the main components of the sample individually, which would be impossible with a technique that does not include a prior separative step such as FTIR.

  1. MIC experimentation needs to be done. Figures are essential to prove the antibacterial effect. 

We find this topic very interesting. Unfortunately, previous studies using this extract showed that its diffusion in Müller-Hinton agar plates is very poor, this limiting the possibility to calculate the MIC using the diameter of the inhibition zones. Instead, we use the broth microdilution tests, followed by the detection of the living cells using a fluorometric measurement method as indicated in Material and Methods. In our opinion, the results obtained using this methodology fit better the definition of MBC rather than MIC.

Minor Comments:

  1. Reference style needs to be corrected.

It has been reviewed and corrected.

  1. Figure quality should be improved. 

Figures have been modified, and hopefully improved.

Reviewer 3 Report

The authors have presented here very simple study. They have focused on the antibacterial and antioxidant activity of grape extract. It would have been great to see if they discover or find out compounds from the extract those are active against tested microorganisms instead. 

From the quantification results it is seen that catechin, procyanidine B1,  and epicatechin  are the major polyphenol present in the extract. I would suggest the authors to test the activity of standards and correlate the results with extract. This way at least readers would know which polyphenols are more effected against tested microorganisms.

The authors are suggested to shorten the introduction. It is too much descriptive and nothing new to readers.

Please check the PDF copy I have marked few corrections in the manuscript as well.

Nothing wrong with English language

Author Response

Reviewer 3 finds that we present a simple study, focused on the antibacterial and antioxidant activity of a grape extract, and would appreciate to know whether individual compounds from the extract are active against the tested microorganisms. He/she suggests testing the activity of standards of the main polyphenols present in the extract and correlating the results with the obtained using the extract and believes that these experiments would allow readers to know which polyphenols contribute the most to the global effect.

Although we appreciate the suggestion of the reviewer, we believe that the antibacterial and antioxidant capacities of the extract are related to the synergism between the different polyphenols present in the extract and not to the individual action of each of them. We think the suggested experiments would not improve the understanding of how the extract works. As we stated in page 14, many studies support this hypothesis. Besides, our objective is to explore the use a complex extract -which is also a by-product of the wine industry- as an alternative to antibiotics or to be used in combination with them, and not to use pure molecules for that purpose.

The authors are suggested to shorten the introduction. It is too much descriptive and nothing new to readers.

We agree. Introduction has been reviewed and shortened as much as possible.

Please check the PDF copy I have marked few corrections in the manuscript as well.

We appreciate the suggestions, and the proper corrections have been done in the manuscript. However, the modification of the TIC using base peak profile is not possible as the software used does not allow such visualization. The TIC shown considers the sum of all the transitions of the method employed.

Regarding the remark of the sentence: “MBC, the lowest concentration”, we assume the reviewer does not agree with the use of MBC instead of MIC. The explanation for this has been given above.

We appreciate the careful evaluation of our work and hope that this revision meets with the reviewers’ approval.

Round 2

Reviewer 1 Report

Acceptable

Author Response

pharmaceuticals-2421152_RESPONSE TO REVIEWERS

The authors thank the reviewer for his/her time and critical reading of the manuscript.

Reviewer 1 considers that the paper is acceptable for publication.

Reviewer 2 Report

Previous comments not complied 

Moderate editing of English language required

Author Response

pharmaceuticals-2421152_RESPONSE TO REVIEWERS

The authors thank the reviewers for their time and critical reading of the manuscript. We have tried to improve it according to what they suggest and have answered their comments on it point by point. We send a new version of the original research article. All corrections in the manuscript are marked using the "Track Changes" option in Microsoft Word.

Reviewer 2 thinks that moderate editing of English is required. In our 1st version of the manuscript this reviewer suggested minor editing of English. We have revised and improved the language of our manuscript and this improvement made the other reviewers consider that the manuscript does not need any further editing now. We do not understand why he/she thinks the English language is now worse than in the first version, since the changes done have been few and apparently for better.

Reviewer 2 also says that his/her previous comments have not been complied. We have already answered the suggestions point by point, saying why we do not agree with some of them:

  1. TFC need to be done

The polyphenolic content has been expressed in terms of TPI (Total Polyphenolic index), which includes both flavonoid content (TFC) and non-flavonoid polyphenols. Both are, in any case, estimated indexes. However, detailed analysis by LC-MS/MS allows not only to identify the main flavonoid and non-flavonoid polyphenols, but also to quantify them. This information, detailed in Table 7, characterizes in detail the polyphenolic profile of the extract.

  1. Qualitative phytochemical analysis needs to be done.

The main components of the extract -and the ones relevant for these study- are polyphenols. Identification and quantification of the main polyphenols have already been done and reported in table 7, as we have explained above.

  1. For antioxidant activities some more experimentation needs to be done e.g. FRAP, H2O2, CUPRAC etc.

Two relevant techniques have already been used. DPPH and ABTS radicals are between the most relevant radicals to detect antioxidant activity. The main objective of the experiments is the evaluation of the antibacterial activity of the extract against clinical samples, although antioxidant activity is an important complementary parameter to describe in depth the nature of the extract. We think that two important techniques are enough to show the high antioxidant capacity of the extract. Furthermore, we remark the great correlation between the results obtained with both techniques.

  1. FTIR need to be done.

FTIR measures the absorption of infrared radiation by a sample and provides information on the functional groups present in the sample. Its main objective is the qualitative characterization of a sample for identification purposes, ideally for pure and/or not very complex substances, highlighting its application to the identification of polymers.

Our extract is a complex mixture, so identification with this type of analytical tool is of little use. For this reason, the identification of the main bioactive compounds is carried out by LC-MS/MS. The first dimension of this technique, liquid chromatography (LC), allows the separation of the compounds prior to their identification. The second dimension, tandem mass spectrometry (MS/MS), allows the unequivocal identification (together with the chromatographic retention time as an additional parameter) of the main components of the sample individually, which would be impossible with a technique that does not include a prior separative step such as FTIR.

  1. MIC experimentation needs to be done. Figures are essential to prove the antibacterial effect. 

We find this topic very interesting. Unfortunately, previous studies using this extract showed that its diffusion in Müller-Hinton agar plates is very poor, this limiting the possibility to calculate the MIC using the diameter of the inhibition zones. Instead, we use the broth microdilution tests, followed by the detection of the living cells using a fluorometric measurement method as indicated in Material and Methods. In our opinion, the results obtained using this methodology fit better the definition of MBC rather than MIC.

Minor Comments:

  1. Reference style needs to be corrected.

It has been reviewed and corrected.

  1. Figure quality should be improved. 

Figures have been modified, and hopefully improved.

Reviewer 3 Report

My suggestion again for the TIC- "change the figure 1 with base peak or mass range option" You don't need sophisticated software for that. You can do it using Xcalibur Qual browser,  and mark the phenolics peaks with numbering.

Author Response

pharmaceuticals-2421152_RESPONSE TO REVIEWERS

The authors thank the reviewers for their time and critical reading of the manuscript. We have tried to improve it according to what they suggest and have answered their comments on it point by point. We send a new version of the original research article. All corrections in the manuscript are marked using the "Track Changes" option in Microsoft Word.

Reviewer 3 suggests the following modification:

My suggestion again for the TIC- "change the figure 1 with base peak or mass range option" You don't need sophisticated software for that. You can do it using Xcalibur Qual browser,  and mark the phenolics peaks with numbering.

Thanks for the comment. Now Figure 2 (SRM reconstructed chromatogram) shows the chromatogram filtered by the quantification transitions of each of the detected polyphenols. Figure 1 shows the TIC obtained with the Xcalibur software. The base peak or mass range option, possible for chromatograms obtained in GC-MS/MS analysis, is not possible in the LC-MS/MS software of our instrument but the peaks corresponding to the polyphenols present in figure 2 have been numbered.